# Exploring the potential of large language models for integration into an academic statistical consulting service–the EXPOLS study protocol

Urs Alexander Fichtner[1,2]*, Jochen Knaus[2], Erika Graf[2], Georg Koch[2], Jörg Sahlmann[2], Dominikus Stelzer[2], Martin Wolkewitz[2], Harald Binder[2], Susanne Weber[2]

1 Faculty of Medicine and Medical Center, Institute of Medical Biometry and Statistics, Section for Healthcare Research and Rehabilitation Research (SEVERA), University of Freiburg, Freiburg, Germany, 2 Faculty of Medicine and Medical Center, Institute of Medical Biometry and Statistics, University of Freiburg, Freiburg, Germany

* Urs.Fichtner@uniklinik-freiburg.de

**Data Availability Statement:** Data collection has not started yet. When data will have been collected, it will be published on a public repository.

## Abstract

### Background

The advancement of Artificial Intelligence, particularly Large Language Models (LLMs), is rapidly progressing. LLMs, such as OpenAI's GPT, are becoming vital in scientific and medical processes, including text production, knowledge synthesis, translation, patient communication and data analysis. However, the outcome quality needs to be evaluated to assess the full potential for usage in statistical applications. LLMs show potential for all research areas, including teaching. Integrating LLMs in research, education and medical care poses opportunities and challenges, depending on user competence, experience and attitudes.

### Objective

This project aims at exploring the use of LLMs in supporting statistical consulting by evaluating the utility, efficiency and satisfaction related to the use of LLMs in statistical consulting from both advisee and consultant perspective. Within this project, we will develop, execute and evaluate a training module for the use of LLMs in statistical consulting. In this context, we aim to identify the strengths, limitations and areas for potential improvement. Furthermore, we will explore experiences, attitudes, fears and current practices regarding the use of LLMs of the staff at the Medical Center and the University of Freiburg.

### Materials and methods

This multimodal study includes four study parts using qualitative and quantitative methods to gather data. Study part (I) is designed as mixed mode study to explore the use of LLMs in supporting statistical consulting and to evaluate the utility, efficiency and satisfaction related to the use of LLMs. Study part (II) uses a standardized online questionnaire to evaluate the training module. Study part (III) evaluates the consulting sessions using LLMs from advisee

**Funding:** This study is financed by the Deutsche Forschungsgemeinschaft (DFG, German Research Foundation) – Project-ID 499552394 – SFB 1597. The recipient of the funding award is Harald Binder. We acknowledge support by the Open Access Fund of the University of Freiburg. The funders had no role in study design, data collection and analysis, decision to publish, or preparation of the manuscript.

**Competing interests:** The authors have declared that no competing interests exist.

perspective. Study part (IV) explores experiences, attitudes, fears and current practices regarding the use of LLMs of the staff at the Medical Center and the University of Freiburg. This study is registered at the Freiburg Registry of Clinical Studies under the ID: FRKS004971.

# 1 Introduction

The progress of Artificial Intelligence (AI), especially in the area of Large Language Models (LLMs) is currently accelerating at an ever-faster pace [1]. LLMs are artificial neural networks trained by using self- and semi-supervised learning. Popular examples are OpenAI's GPT models, Google's PaLM, DeepMinds Gemini and Meta's LLaMA and Microsoft's KOSMOS [2].

LLMs become more and more attractive in various contexts of the scientific process, e.g. supporting the production of scientific texts, knowledge synthesis, translation–but also data analysis. In the medical context, the use of LLMs has especially been examined in order to support the communication with patients [3–5]. Not only in the context of medical research, the integration of LLMs is a cutting-edge advancement that is revolutionizing research across various fields. Garrel et al. examined the use of AI at a higher education landscape via a survey among students in Germany [6]. LLMs can take over "roles", e.g. those of programmers or data analysists using statistical software, e.g. Python or R, and can answer to questions that deal with problems of data analysis. It is notable that LLMs have the capability to perform tasks beyond their original intended purposes. Dell'Acqua et al. investigated how access to ChatGPT 4 impacts productivity and quality of consultants within a global management consulting company [7]. LLM tools can act as "statistical consultant" supporting to find answers and solutions for analysis strategies, programming issues and bug fixing. In order to evaluate the quality of statistical programming code generated by ChatGPT, Liu et al. conducted a systematic analysis of 4,066 ChatGPT-generated code snippets for 2,033 programming tasks in Java and Python [8]. They found that in every fourth case, the produced output was wrong. Their experiments suggest that ChatGPT showed potential for self-repairing of code, however they conclude that it still has limitations in its current state. Their study evaluated the potential of ChatGPT 3.5 which was trained with much less data than its successor ChatGPT 4. The newest GPT model of OpenAI was released in March 2023 and was trained with 175 billion training parameters. Furthermore, a new plugin, called "Advanced Data Analysis" (ADA) was released for ChatGPT 4 that provides specific features for data evaluation. After an update, this feature is now available as GPT called "Data Analyst". With this feature, users can upload their data and let the system analyse them by using textual questions (called "prompts"). ADA introduces an extensive range of new application areas. For instance, Müller et al. investigated opportunities and limitations of this plugin for hydrological analysis [9]. However, both the use of LLMs as "consultant", as well as "data analyst" face several risks [10,11]. First, the quality of the outcomes of LLMs has to be verified and might depend on several aspects, as e.g. the quality of training data, the difficulty of the question and the wording of the prompts. Second, the quality of the outcomes needs to be validated and verified and it stands to question whether the outcome quality can be judged in a general manner. As long as the outcome quality and precision of the operations of the LLM is not guaranteed, it might produce errors of different levels of severity. Third, without deeper understanding and education, the advice-seeking person might not be able to correctly differentiate between invalid and proper answers or analysis results. Fourth,

uploading data onto a server can be problematic for reasons of data protection and limits the use of ADA. Ignjatovic et. al investigated the use of ChatGPT 3.5 and 4 in order to solve biostatistical problems. They conclude that with ChatGPT 4 a good level of performance can be obtained, but the usage of such tools for solving biostatistical problems should be done with caution [12].

At the Medical Faculty of the University of Freiburg, statistical consulting is offered to every member of the faculty who searches supervision or support of questions related to data analysis. In some exceptional cases, even the whole analysis process is executed by the statisticians employed there. Since many medical researchers have limited education in data science, this consultation is offered to support good scientific practice using adequate analysis methods by well-trained field experts. Depending on the statistical expertise of the advisees and the complexity of the analysis questions, the consultation offer varies between a unique e-mail-based advice over two or three face-to-face or web-conference consulting sessions to possibly even programming meetings in presence of the advisee. In order to sustain high quality in statistical consulting, a standardized training and orientation concept has been developed for general quality improvement. This concept has been mandatory to be absolved for consultant beginners since 2019. Additionally, regular bi-weekly meetings are held for collegial exchange, where consultation cases are presented for joint discussion of consulting practices.

It is foreseeable that LLMs will play a role in the future handling of data analysis and thus, in the procedures that are aligned to statistical consultation. Both advice-seeking scientists and statistical field experts might make use of such tools to support their work. We anticipate that, in a best case scenario, the use of a LLM will enable consultants to significantly improve their efficiency in managing consulting tasks. Consultants could benefit from a shift of their working duties from programming tasks to more conceptual thinking tasks. Published studies point out this potential, however, successful support in terms of being more efficient when performing data analysis tasks using LLMs still stands to question [8,13]. Critical output assessment and learning in how to formulate prompts seems to be crucial.

The question of how to integrate LLMs in research is, however, not limited to the medical field. LLMs can be used for text and code production and thus, shows potential for all research areas, including teaching. Rahman & Watanobe addressed opportunities, threats, and strategies of ChatGPT for education and research [14]. However, many contextual factors might play a role in order to determine a qualified use of LLMs with high efficiency, utility and user satisfaction. Those factors might lie in the persons interacting with the systems, e.g. level of education in statistics, competence and experience in using LLMs, attitudes and fears towards LLMs and personal preferences. Additionally, also factors within the technology used might play an important role, e.g. the outcome quality of the LLM, its interoperability, its flexibility, its transparency, the repeatability of the outcomes as well as its comparability and ease of use. Gärtner et al. investigated the use, benefits, and limitations of ChatGPT and Large Language Models (LLMs) in teaching at Bavarian universities of applied sciences [15]. They found that the usage of LLMs in teaching, studying and daily professional life is still low due to the lack of application possibilities, poor quality of results and legal concerns of the potential users. On the other side, respondents viewed the use of LLMs for teaching as an opportunity and expressed the need for support by their employers.

## 1.1 Statistical consulting

Many statistical research organizations offer methodological support to clinical and life science researchers. This may happen in individual consultations without further involvement, in project-specific co-operations or in long-term collaborations among domain experts and

biostatisticians [16]. Consultation can be provided on aspects of study design and methodological evaluation, and the advice given can refer to basic research or to clinical research projects.

At the Institute of Medical Biometry and Statistics (IMBI), a free consulting service is operated by a cross-section of all employed statisticians and mathematicians, with more than 20 IMBI scientists providing consulting in addition to their research activities. Over 250 requests for advice are processed each year. Those seeking advice are members of the Medical Faculty of the University of Freiburg or the Freiburg University Medical Center. The advisees range from students with questions about doctoral projects and postdocs to experienced research group leaders and principal investigators. The contents addressed range from study planning including sample size calculation and planning of the statistical analysis in a study protocol through application of statistical methodology in a given data set.

At the IMBI, and hence within the EXPOLS project, consulting services are defined as follows:

"Advising in a single or a few advising session(s) (sometimes called consulting service), where the advisee poses questions on statistical aspects in a particular project and the adviser discusses possible solutions but does not further engage in the project." [16]

In particular, no further engagement means that no data analysis is performed by the consultant. As there is a wide range in the research areas, research questions and statistical expertise of advisee, there is a huge heterogeneity between the consulting sessions.

## 1.2 The integration of a LLM into statistical consulting

At the IMBI, statistical consulting is partly standardized. Every advisee fills out a template providing information on the project and their requests. Then, the administrative team of the statistical consulting group assigns the case to a consultant based on the best matching expertise. How the consultation is operated is not prescribed and varies from case to case and from consultant to consultant. Usually the consultants get in touch with the advisees and, if necessary, meet with them (online). To enhance the advice provided by consultants, retrieving information from literature or utilizing other informative materials from the internet can be instrumental. Depending on the request and the expertise of the consultant, more or less preparation is necessary. In some cases, written feedback is also sufficient. In the context of this study, each statistical consultant has received access to ChatGPT 4 and is advised to integrate the LLM in the consultations. Due to the considerable heterogeneity among consulting sessions, the tasks for which the LLM can be used will also vary. Potential uses of the LLM include pre- or post-session preparation, real-time assistance, data analysis guidance, drafting documentation, serving as an educational tool, and many more. Consultants are not instructed to use ChatGPT in a specific manner or for specific tasks and possible scenarios could be that the consultant integrates the LLM as a third person during the consulting session, that the consultant uses it for producing analysis code, that the consultant uses it for summarizing scientific texts, and so on. One aim of this project is to assess how this integration works and how it can support statistical consulting.

## 2 Materials and methods

### 2.1 Research questions

Within this project, we aim to answer the following research questions:

1. How can LLMs be used to support statistical consulting?

1.1 How do advisees and consultants perceive and rate the use of LLMs in statistical consulting?

1.2 What are the strengths, limitations and areas of improvement in the use of LLMs for statistical consulting?

1.3 What factors play a role in determining the use of LLMs with high efficiency, utility and user satisfaction?

2. Is the developed training module for statistical consultants and (future) scientists efficient and sustainable, as measured by the quality and relevance feedback from the participants?

3. What are the experiences, attitudes, fears, beliefs and current practices regarding the use of LLMs?

## 2.2 Primary and secondary outcomes

This study is divided into four sub-studies using different methods to gather data. Each sub-study addresses one or more research questions and thus has its own measurement goals und study population (see section Study population). However, as this is an exploratory study, no hypotheses will be formulated in advance and thus, no clearly prioritized outcome measures can be derived. The outcome measures are displayed by study part in subsequent Table 1.

The schedule of enrolment, interventions and measurements covering all measurement time points for the study samples according to the SPIRIT guideline is displayed in Fig 1.

## 2.3 Study design

This monocentric study will utilize qualitative and quantitative methods to explore the use of LLM in an academic statistical consulting service from both consultant and advice-searching staff. Each of the four sub-studies addresses different research questions and is organized in working packages (WP) (see Table 2).

WP 1: To develop the training module, existing teaching material from online resources, prior course visits and own preparatory works will be consolidated. The course consists of a lecture part and a hands-on part. The whole teaching session is estimated to take approximately 2–3 hours. Sustainable reuse potential of the course will guide the development of the training module.

WP 2: The course will be offered in a synchronous hybrid format to enable participants to take part from home office. We offer up to three different dates to ensure that every statistical consultant can take part in the training session. Participation will be mandatory as part of consultants' work-related duties. The course will be offered to all consultants employed at the IMBI and is not restricted to those who participate in the study. The course will be conducted by SW, the principle investigator of this project.

WP 3: For all study parts, we will make use of standardized online questionnaires implemented via REDCap. Those questionnaires will be constructed based on existing literature and earlier used items and scales in other, comparable contexts. The questionnaires will be offered in bilingual form (GER, ENG). The interview guideline applied in study part I will be developed within a consortium of field-experts from Sociology, Empirical Social Science, Statistics, Health Services Research, Medicine and Mathematics.

WP 4: Before the training session, we will ask consenting statistics consultants to fill out a short questionnaire provided via REDcap on their expectations about the use of LLM in

**Table 1. Study outcomes.**

| Study part | Population | Outcomes | Research question |
|---|---|---|---|
| I: Mixed mode study using qualitative and quantitative methods (online, standardized and online, semi-structured) | Statistical consultants (Institute of Medical Biometry and Statistics) | Quantitative: Result expectation Result achievement Satisfaction with use of LLM Quality Assessment Communication Assessment Efficiency Assessment Future potential Perceived risks and limitations Qualitative: Result expectation Perceived barriers Perceived potential Possible pitfalls and solutions Experiences | 1, 1.1, 1.2, 1.3 |
| II: Quantitative standardized online questionnaire | Statistical consultants (Institute of Medical Biometry and Statistics) | Appropriateness of the time frame of the training module Structure of the content of the course Perceived value of content Perceived learning success Quality in communication Motivation of the course Preparation quality of the course Willingness to recommend the course Use of training session for future consulting tasks Strengths and limitations, improvement potential Overall quality rating of course | 2 |
| III: Quantitative standardized online questionnaire | Advisees (Clinicians and domain experts) | Evaluation of the absolved consulting session (captured via regular evaluation form as a score and categorised) Satisfaction with the use of LLM in consultation Quality assessment Communication assessment Efficiency assessment | 1.1 |
| IV: Quantitative standardized online questionnaire | Scientific employees (researchers and clinician scientists at the University Medical Center and the Albert-Ludwigs-University Freiburg) | Attitudes and beliefs towards the use of LLM Experiences with the use of LLM Current usage of LLM and contexts Perceived benefits in using LLM Perceived barriers in using LLM Knowledge of or participation in development of guidelines / rules towards the use of LLM Changes in practice in line with the introduction of LLM Differences across faculties and research fields | 3 |

statistical consulting. Furthermore, we will collect data on potential predictors that might have influence on the adaptation and use of LLM in statistical consulting.

WP 5: With a purposive sampling approach, we will select consultants and will conduct semi-structured guideline-based qualitative interviews to gather in-depth insights into their perceptions regarding the use of LLM in statistical consulting, the associated barriers, limitations and potential benefits. The interviews will be conducted via bilateral Webex-Meetings. The interviews will be digitally recorded.

WP 6: All interviews of WP 5 will be transcribed verbatim and analyzed qualitatively by SW and UF following a parallel coding approach according to Kuckartz [17]. The results will be used to refine survey instruments that will be used in WP 7 and 9.

| TIMEPOINT** | $t_0$ | Measurement $t_0$ | Teaching (Intervention) | Measurements | | |
|---|---|---|---|---|---|---|
| | Informed Consent | | | $t_1$ | $t_2$ | $t_3$ |
| Study part I (Statistical Consultants) | X | X | X | X | X | X |
| Study part II (Statistical Consultants) | X | | X | X | | |
| Study part III (Advisees) | X | | | ◆———————————◆ | | |
| Study part IV (University and medical center staff) | X | | | X | | |

**Fig 1. Schedule of enrolment, interventions, and measurements.**

WP 7: After all consultants participated in the training session, they are asked to actively involve LLM in their subsequent consultation sessions. At t1 (one week after training session), t2 (two months after training session) and t3 (6 months after after training session), the consultants will receive a standardized online questionnaire provided via REDcap.

**Table 2. Working packages and time table.**

| Working package (WP) | Month | | | | | | |
|---|---|---|---|---|---|---|---|
| | I | II | III | IV | V | VI | VII |
| WP 1: Development of training—course conceptualization | ▓ | | | | | | |
| WP 2: Training session with all statistical consultants | | ▓ | | | | | |
| WP 3: Development of questionnaires, instruments, interview guideline | ▓ | | | | | | |
| WP 4: Pre-training survey among consenting consultants | ▓ | | | | | | |
| WP 5: Semi-structured qualitative interviews with consultants | | ▓ | | | | | |
| WP 6: Analysis of qualitative interviews and synthesis of results | | ▓ | ▓ | | | | |
| WP 7: Post-training survey among consultants (T1: one week after training, T2:three months, T3: six months) | | ▓ | | | | ▓ | ▓ |
| WP 8: Continuous post-consulting survey among advisees | | ▓ | ▓ | ▓ | ▓ | ▓ | ▓ |
| WP 9: Survey among all members of the Medical Faculty / Medical Center | | | ▓ | | | | |
| WP 10: Analysis and preparation of findings, knowledge synthesis | | | | | ▓ | | |
| WP 11: Dissemination of results, data and evaluation of need for further research (Study part II, IV) | | | | | | ▓ | |
| WP 12: Analysis and preparation of findings for follow-up (Study part I, III) | | | | | | | ▓ |

WP 8: All advisees participating in the study will be asked to fill out an evaluation questionnaire after their consultation session. This questionnaire is attached as a link to the regular evaluation of the statistical consulting session and is executed as a separate REDCap form. The questionnaire also captures whether or not a LLM was used in the consulting session and what the reasons for non-use were. The survey will also contain questions regarding the advisee's perception towards the use of LLM during the session.

WP 9: Scientific employees at the University Medical Center and the Albert-Ludwigs-University Freiburg will be asked to complete a standardized online questionnaire provided via REDCap on a voluntary basis. For recruiting, we will use several different methods to ensure reaching all staff involved in research. The questionnaire will be online for 6 weeks.

WP 10: The baseline data of study parts I to IV will be analyzed exploratory and in a descriptive manner. Potential correlations or group-differences will be tested bivariately.

WP 11: The quantitative data and the results will be published in adequate scientific journals (see data publication) and needs for further research will be evaluated.

WP 12: The follow-up data of study parts I and III will be analyzed exploratory and in a descriptive manner. Results will be published in adequate scientific journals.

A detailed overview of the study recruitment workflow is illustrated in the Section "Data management plan".

## 2.4 Study population

The study population varies and thus is described for each study part.

**Study part I (Mixed mode study using qualitative and quantitative methods (online, standardized, semi-structured)):**

Inclusion criteria: All potential study participants must be members of the statistical consulting team at the time of data collection and have signed a declaration of informed consent. The statistical consulting team consists of 24 consultants. Assuming a participation rate of 80–90% the estimated sample size for the quantitative part is n = 20; estimated sample size for the qualitative part: n = 6.

Due to the interest already expressed among the consultants, the assumed participation rate seems realistic. We expect a high level of intrinsic motivation based on professional involvement. Participation can take place during regular working hours so that no additional private time resources need to be invested. In order to ensure a high participation rate we implement repeated reminders.

**Study part II (Quantitative standardized online questionnaire):**

Inclusion criteria: All potential study participants must be members of the statistical consulting team at the time point of data collection and have given electronic informed consent. Similar to study part I, the estimated sample size is n = 20.

**Study part III (Quantitative standardized online questionnaire):**

Inclusion criteria: All potential study participants must have absolved at least one statistical consulting between the date of the teaching of the consultants and the end of the observation period. Electronic informed consent needs to be obtained in advance. There are at least 15 consultations per month. With a data collection period of 6 months this results in at least 90 consultations. Assuming a participation rate of 33% the estimated sample size is n = 30.

**Study part IV (Quantitative standardized online questionnaire):**

Inclusion criteria: All potential study participants must be employed as scientific or medical staff at the University of Freiburg or the University Medical Center or must work closely with

science. The declaration of consent is integrated into the start of the online survey. The source population consists of approximately 5,800 scientific employees. In addition, employees working closely to science, but not officially declared as scientists, are also invited to participate. Assuming a response rate of 10–20% the estimated sample size is 600–1,200.

Exclusion criteria: All members that do not fulfil the inclusion criteria will be excluded. Furthermore, potential participants will be excluded if they do not have sufficient language skills in either German or English.

## 2.5 Recruitment

Study participants for study part I and II are employed in our institute and will be contacted via their associated company e-mail address. Project members are part of the study population and can participate both in the qualitative and the quantitative part. The principal investigators have no authority to issue directives within the consulting team. There are no conflicts of interest, as participation in the study is voluntary and no supervising dependencies are apparent. Participation or non-participation in the study has no impact on the employment. To ensure this, the questionnaires are treated confidential.

For the qualitative study, a purposive sampling approach will be chosen and participants will be recruited by personal contact.

Study participants for study part III will be recruited via the statistical consultant.

Study participants for study part IV will be recruited via multiple media, e.g. newsletters, as well as newsfeeds, e.g. of the University Medical Center.

## 2.6 Instruments

The following data / instruments will be collected / used (Table 3).

## 2.7 Risk-benefit assessment

Individual benefit associated with study participation:

Study part I, II: The study participants are trained in the use of LLMs and can therefore develop a differentiated opinion on integrating modern AI tools for their work.

Study part III: Study participants learn about potential benefits and risks for their own methodological data analysis under expert guidance.

Study part IV: There is no individual benefit to participating in the anonymous survey.

Burdens and risks associated with study participation:

Study part I, II: Familiarization with new tools is always necessary for scientific staff. The additional workload is minimal due to potentially longer counselling sessions.

Study part III: there is a risk of potential minimal increased time effort in statistical consultation.

Study part IV: no risk.

## 2.8 Sample size

Since this study is of explorative nature and we aim to conduct a survey with the full basic population, sample size calculation is not applicable.

1. Study part I, II: statisticians (consultants): the sample size planning is based on the current consultants (N = 24). Assuming a participation rate of 80–90%, we expect data of n = 20 statisticians.

**Table 3. Instruments used for EXPOLS.**

| Study Part | Data / Instrument |
|---|---|
| I | Experience as consultant<br>Sex<br>Typical consultation format<br>References used for consultation<br>Experience with use of AI for statistics<br>Usability of AI (UMUX; [18]) covering effectiveness, efficiency and satisfaction<br>Technology Readiness (TRI 2.0; [19])<br>• Optimism (4 items)<br>• Innovation (4 items)<br>• Discomfort (4 items)<br>• Insecurity (4 items)<br>Result expectation [20]<br>• Performance Outcome Expectation (POE) (3 Items)<br>• Self-Evaluative Outcome Expectation (SEOE) (3 Items)<br>• Social Outcome Expectation (SE) (3 Items)<br>Perceived benefits and disadvantages in using AI<br>Usage of ChatGPT after training session<br>Reasons for non-usage |
| II | Learning success<br>Competence gain<br>General teaching competence<br>Potential use of ChatGPT<br>Preferred teaching mode<br>Overall rating<br>Recommendation potential: Net Promoter Score [21] |
| III | Use of AI during consultation<br>Satisfaction with use of AI during consultation<br>Usability of AI (UMUX; [18])<br>Perceived benefits and disadvantages in using AI |
| IV | Status group<br>Experience in science<br>Faculty membership<br>Sex<br>Experience using AI: areas<br>Potential use of AI in the future: areas<br>Difficulties using AI<br>Technology Readiness (TRI 2.0; [19])<br>• Optimism (4 items)<br>• Innovation (4 items)<br>• Discomfort (4 items)<br>• Insecurity (4 items)<br>Result expectation [20]<br>• Performance Outcome Expectation (POE) (3 Items)<br>• Self-Evaluative Outcome Expectation (SEOE) (3 Items)<br>• Social Outcome Expectation (SE) (3 Items)<br>Experience and potential for training sessions with the topic AI<br>Experience in using AI for teaching<br>Fears about the role of AI in the future<br>Knowledge, use and need for guidelines regarding AI<br>Use of AI in the patient-provider relationship (only medical staff) |

2. Study part III: domain experts (advisees): considering January until May 2023 as reference over 15 consultations per month can be expected. With a duration of 6 months for data collection of this subpopulation and an expected response rate of 33%, we expect data of n = 30 domain experts (advisees).

3. Study part IV: scientific employees: 2,623 scientists were employed 2022 at the University and 3,189 persons were employed in 2022 with association to the Medical Faculty working

in science. In addition, employees working closely to science, but are not officially declared as scientists, are also invited to participate. Assuming a response rate of 10–20% the estimated sample size is 600–1,200.

## 2.9 Statistical methods

Analyses will be explorative.

**Qualitative analysis:**

Qualitative data from study part I will be transcribed verbatim and then will be analyzed with the structuring content analysis approach [17].

Free-text responses from study part II-IV will be thematically analyzed to identify common themes and impressions.

**Quantitative analysis:**

Data collected through Likert-scale or multiple-choice questions will be described using descriptive statistics. Absolute frequencies will be provided for the number of consultations and consultants. The number of evaluations will be presented for both consultations and consultants (absolute and relative). Ratings of consultations through the evaluation form will be described using arithmetic mean with a 95% confidence interval, standard deviation, minimum, 25th percentile, median, 75th percentile, maximum, and the count of complete and missing values.

For the pre-post comparison of ratings, Mann-Whitney U tests will be employed, since we expect a small sample size. In this case, the distributive assumptions for parametric approaches will probably be violated and non-parametric procedures are favorable [22,23].

## 2.10 Data management plan

The study recruitment workflow is illustrated in the Figs 2–4.

**General data in statistical consulting and use of the "ChatGPT" service (OpenAI):**

As a matter of principle, the consultants do not accept transmission of any research data as part of the statistical counselling at the IMBI; for the majority of cases the activity is purely consultative. However, the consultants are free to enter into a cooperation project with the advisees if they are interested in methodological issues or for other scientific reasons. For data protection reasons, this cooperation is not covered by the general regulations of the statistical consulting and therefore must be evaluated and documented separately as a normal scientific project.

This means that the data on which the consulting activities are based are not available at the IMBI and are not transferred to the Institute.

In the context of the present study, a LLM supports the consulting activities and is subject to the restrictions of statistical consulting, so no data is transferred to the IMBI, stored or processed at the IMBI. There are certain LLMs that can be used in local data centers (e.g. Llama), but these do not yet offer the range of functions of ChatGPT and the Advanced Data Analyses (ADA) function in particular can currently only be used here. The ChatGPT language model used in the study can only be used as a commercial service from a non-European company (OpenAI) in an American data center. The establishment of third-country processing with the conclusion of an order data processing contract with correspondingly extended standard contractual clauses is not intended. This means that the transfer of data relevant to data protection law, i.e. generally clinical study or routine data, to the ChatGPT web service is not permitted within the scope of this study.

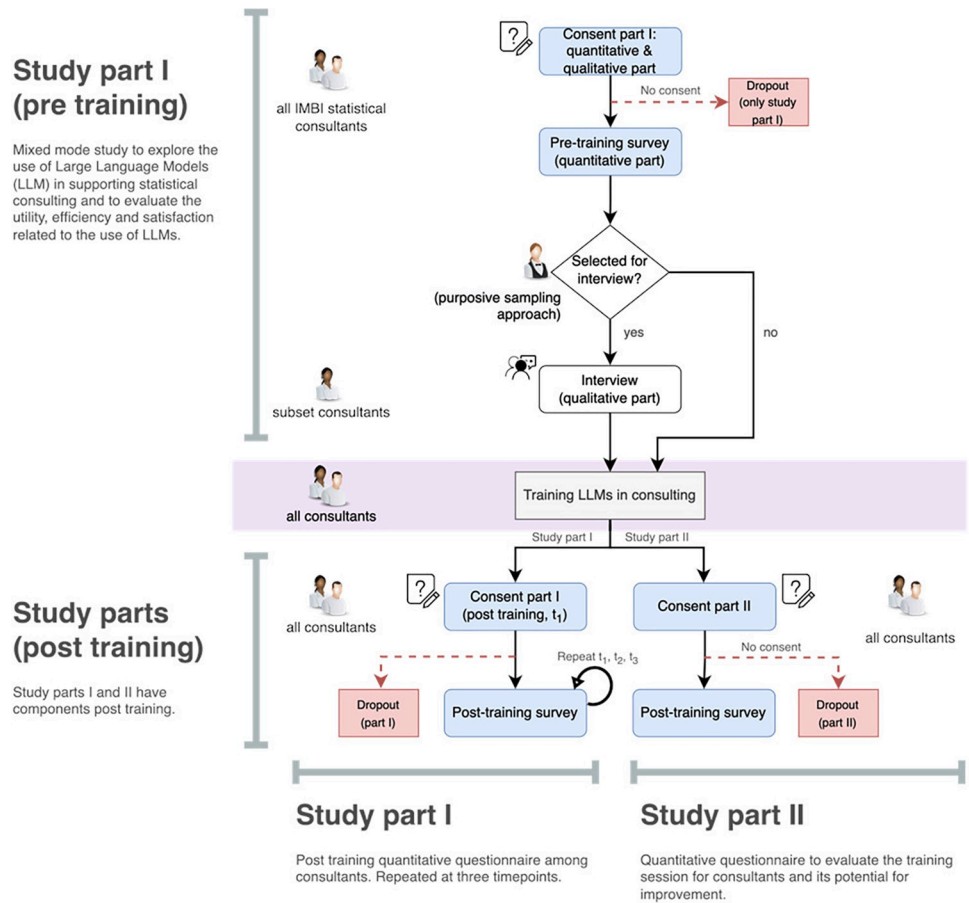

**Fig 2. Recruitment scheme study parts I & II.**

The creation of simulation data, which may be transferred for the use of ADA, should be generated under the guidance of the consultant–if ChatGPT is used for simulation data creation, no original patient data must be used. All statisticians are obliged to comply with the standards of the University Medical Center and try to sensitize the advisees so that data is not imported into ChatGPT or other services under data protection regulations outside the counselling process.

## Survey data (quantitative data)

Detailed information and procedures are described in the attached data protection concept.

**Study part I-IV:** The questionnaires are collected via the REDCap data capture report form system. It is provided by the Clinical Trials Unit (ZKS Freiburg). Technically, the public instance is used, which allows the questionnaires to be used outside the University Medical Center network (essential for parts III and IV with potential participants from pre-clinical institutes and the University). Analysis data sets (cleaned questionnaires) are transferred as CSV files to a network drive of the University Medical Center. Access is restricted to the study evaluators.

**Study part I, II:** Due to the small number of cases and the known participants (statistical advisory team of the IMBI), there is an increased risk of combinatorial re-identification of using the quantitative data. Here, specific information such as "years of professional

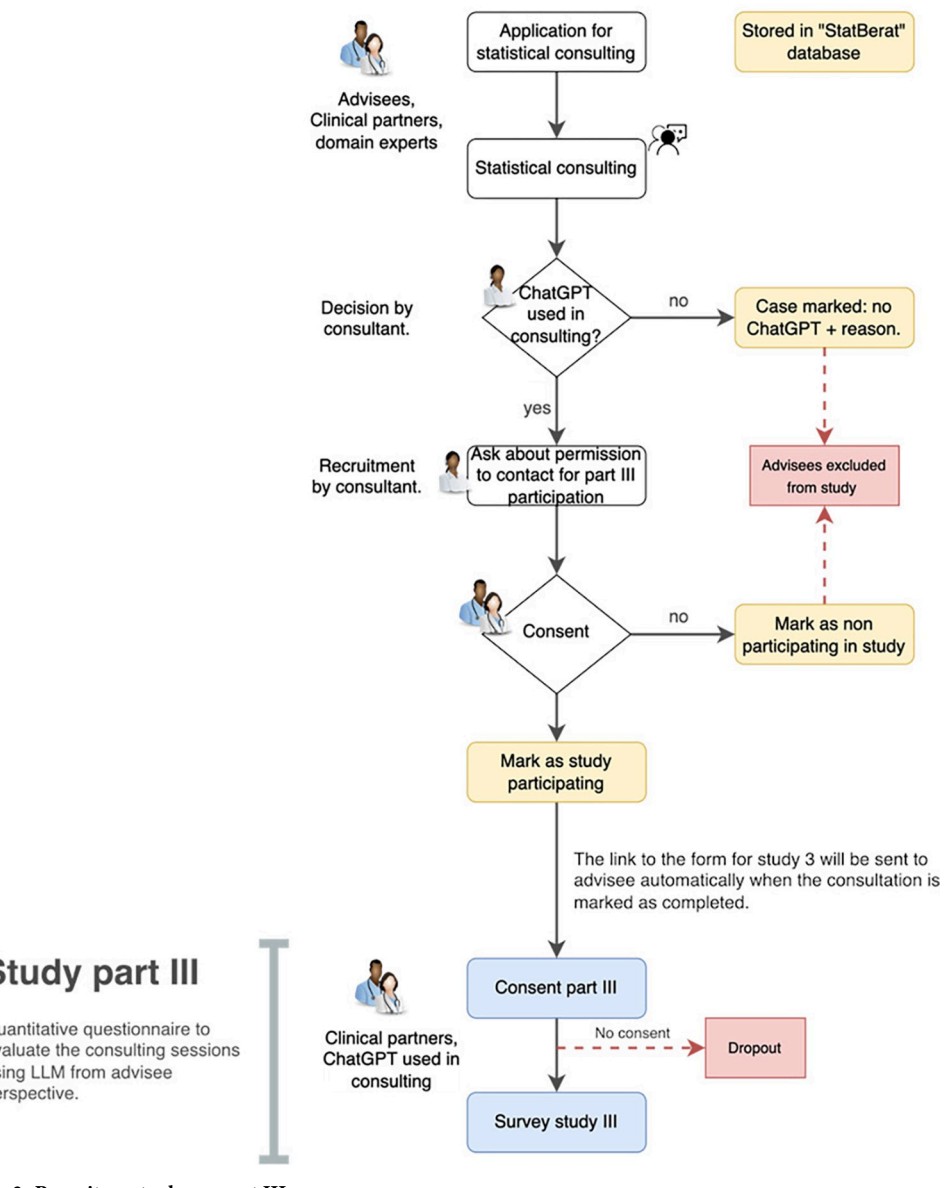

**Fig 3. Recruitment scheme part III.**

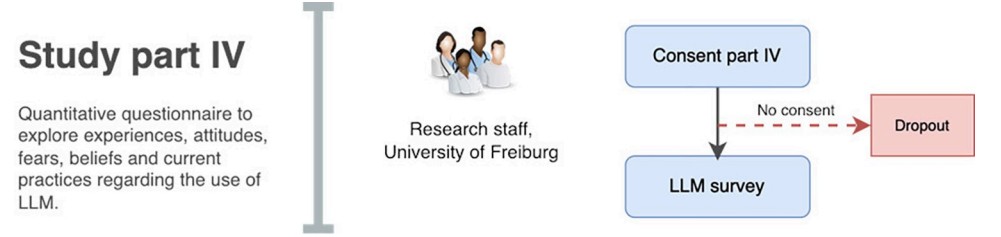

**Fig 4. Recruitment scheme part IV.**

experience" is already sufficient for re-identification. This means that the quantitative data sets are excluded from publication and the questionnaire is designed in such a way that it can be used without any consequential risk.

### Survey data–pseudonymization/anonymization

Study part I: participants will be pseudonymized to allow for contact for interviews.

Study part II: participants will be anonymous in the survey (not in the qualitative part).

Study part III: surveys will be pseudonymized in order to combine the study response with the evaluation response of the overall consultation. Only study leaders have access to the pseudonymization list.

Study part IV: participants are anonymous in the survey.

Free text fields will be cleaned before evaluation: any potentially identifying information will be stripped and information will be categorized where specific terms could lead to additional risk of re-identification.

### Consent

This study is registered at the Freiburg Registry of Clinical Studies under the ID: FRKS004971. The ethics committee of the University of Freiburg has approved the study on 11 June 2024 (24-1148-S1).

Electronic informed consents will be obtained for all parts of a multi-part study. For part III, verbal consent will be given for contact during consultations, and questionnaire-related consents are strictly managed, with applications to statistical consulting anonymized one year post-consultation and retained for two years post-study.

### Audio files

Study part I: interviews will be recorded for later transcription. This transcription will be done during the study period. The interview data will be stored on a University Medical Center network drive and is only accessible to the study evaluators.

### Data publication

In line with good scientific practice, the data will be published in public repositories as far as possible.

Study part I-IV: all free text fields will be cleaned and categorized. Sensible information in free text will be removed at all.

Study part I, II: no data will be published.

Study part III: study data will be published, but without the connected information on the statistical consulting case.

Study part IV: study data will be published for reuse of the scientific community.

### Status and timeline

In line with the working packages displayed in Table 2, a timeline of 7 months is expected. At the time point of submission of this study protocol, the digitalization of questionnaires for study part I and IV is executed. Furthermore, a concept of the training session is finalized and ready for application. No data was collected so far.

## 3 Discussion

With this study, we expect to gather insights into the potential integration of LLMs into statistical consulting using the example of ChatGPT. By this, we are able to investigate the strengths and limitations of using LLMs for statistical consulting in an academic environment and can identify potential usage patterns of our study population. To our knowledge, this is the first study dealing with this topic using different types of data. Furthermore, we aim to explore experiences, attitudes, fears and current practices regarding the use of LLMs of the staff at the Medical Center and the University of Freiburg. The strength of this study concept is that it combines qualitative and quantitative methods and implements them from different perspectives (statistical consultants, advisees and staff of the University and Medical Center). This enables a triangulation of data and might contribute to a better understanding of the studied field. This approach offers the possibility to compensate for the shortcomings of using only either qualitative or quantitative methods and is in line with the Mixed-Methods principle. By approaching our research questions using different methods, we expect to reach for the broad purposes of breadth and depth of understanding and corroboration [24].

However, the study design is limited in its generalisability. It is unclear whether the setting of statistical consulting at other college and university organizations is similar to the IMBI. Depending on the response rates, we can at least draw conclusions based on surveying the full basic population at Freiburg. Furthermore, this small basic population limits potentially applicable statistical methods, especially if response rates will be lower than expected. Our high expected response rates lie within the anticipated importance of the topic and feedback we received when planning the study. However, it is debatable how realistic these assumptions are. We will take all non-monetary measures, e.g. reminders and broad communication of the survey parts via various media, to increase the response rate. Especially study part IV might benefit in its representativeness from rolling out the survey across different Universities on national or even international level. Within the given timeframe, this was not executable. The conceptualization of a cross-institutional study would require a longer preparation period with external funding. However, we will publish all material used for potential replications in the future. By using administrative data on professions and numbers of employees, we can compare our collected data with the real distribution at the University and Medical Center of Freiburg and are able to draw conclusions about coverage.

## Supporting information

**S1 Checklist. SPIRIT 2013 checklist: Recommended items to address in a clinical trial protocol and related documents\*.**
(DOC)

**S1 File. Original study protocol.**
(PDF)

## Author Contributions

**Conceptualization:** Urs Alexander Fichtner, Jochen Knaus, Erika Graf, Jörg Sahlmann, Dominikus Stelzer, Susanne Weber.

**Data curation:** Urs Alexander Fichtner, Jochen Knaus, Susanne Weber.

**Formal analysis:** Urs Alexander Fichtner, Susanne Weber.

**Funding acquisition:** Harald Binder.

**Investigation:** Urs Alexander Fichtner, Susanne Weber.

**Methodology:** Urs Alexander Fichtner, Susanne Weber.

**Project administration:** Urs Alexander Fichtner, Jochen Knaus, Georg Koch, Susanne Weber.

**Resources:** Jochen Knaus.

**Software:** Urs Alexander Fichtner, Georg Koch.

**Supervision:** Erika Graf, Martin Wolkewitz.

**Validation:** Urs Alexander Fichtner, Susanne Weber.

**Visualization:** Jochen Knaus.

**Writing – original draft:** Urs Alexander Fichtner, Martin Wolkewitz, Susanne Weber.

**Writing – review & editing:** Jochen Knaus, Erika Graf, Jörg Sahlmann, Dominikus Stelzer, Harald Binder.

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
