## [Decision Letter · Decision Letter 0]

16 Jul 2024

PONE-D-24-24617Exploring the Potential of Large language models for integration into an academic Statistical Consulting Service – The EXPOLS Study ProtocolPLOS ONE

Dear Dr. Fichtner,

Thank you for submitting your manuscript to PLOS ONE. After careful consideration, we feel that it has merit but does not fully meet PLOS ONE’s publication criteria as it currently stands. Therefore, we invite you to submit a revised version of the manuscript that addresses the points raised during the review process.

We look forward to receiving your revised manuscript.

Kind regards,

Bekalu Tadesse Moges

Academic Editor

PLOS ONE

Journal Requirements:

4. We note you have included a table to which you do not refer in the text of your manuscript. Please ensure that you refer to Table 3 in your text; if accepted, production will need this reference to link the reader to the Table.

Additional Editor Comments:

Dear Author(s), You need to consider improving the quality of the protocol, as suggested by the reviewers.

1. It is rationalized that training material to integrate LLMs into statistical results is needed. However, the second research question related to the development of such material is formulated. I think the material needs to be evaluated, based on the data collected, in terms of its quality and relevance for the advisee and consulting jobs. Thus, I suggest this very question needs revision.

2. if the project needs mixed-methods study, it is better to choose or develop a mixed methods design as appropraite, that fits into your study conceptualization. the mixed methods should allow genuine integration of both qualitative and quantitative data (Teddlie and Tashakorri, 2009, Creswell, 2020)

3. Why is the Mann-Whitney U test selected? it is a non parametric equivalent to the independent samples t-test, used when data doe not meet the assumptions of parametric test. please look into it. (Note that, If you argue otherwise to the comments, they should be convincing, and you are welcome)

Reviewers' comments:

Reviewer's Responses to Questions

**Comments to the Author**

1. Does the manuscript provide a valid rationale for the proposed study, with clearly identified and justified research questions?

Reviewer #1: Partly

Reviewer #2: Yes

2. Is the protocol technically sound and planned in a manner that will lead to a meaningful outcome and allow testing the stated hypotheses?

Reviewer #1: Partly

Reviewer #2: Yes

3. Is the methodology feasible and described in sufficient detail to allow the work to be replicable?

Reviewer #1: Yes

Reviewer #2: Yes

4. Have the authors described where all data underlying the findings will be made available when the study is complete?

Reviewer #1: Yes

Reviewer #2: Yes

5. Is the manuscript presented in an intelligible fashion and written in standard English?

Reviewer #1: Yes

Reviewer #2: Yes

6. Review Comments to the Author

You may also provide optional suggestions and comments to authors that they might find helpful in planning their study.

Reviewer #1: Major Points of Review

1. The title is interesting and ideal, make it summarize the specific

Part 1: Introduction

1. The introduction does not clearly articulate the problems of data analysis or the rationale for this specific study regarding the Potential of Large language models for integration into an academic Statistical Consulting Service. For example, it should explain the problem of data quality, the problem of statistical methods, and the gap of statistical software or another specific purpose. The novelty of large language models should be articulated, highlighting any advancements.

Part 2: The integration of an LLM into statistical consulting.

1. The integration of LLM into statistical consulting is not well explained for the pre-session preparation, data analysis guidance, and real-time assistance as well as assessing integration.

2. Some of the research questions are not measurable. For instance, how to measure, efficiency, utility, and user satisfaction?

Study Design

1. Why incorporate qualitative methods, b/c more of such types of scenarios are more appropriate with quantitative methods if it is possible, please clearly articulate the reason for why.

2. The qualitative and quantitative methods are different aspects, so how do design at a time?

3. Is the given study population representative?

Part 3: Statistical methods

1. Qualitative and quantitative analysis is much more limited, regarding data type, purpose focuses, and others will not provide.

Part 4: Novelty and Contribution

The paper claims to contribute to improving the LLM. The novelty of this contribution is not fully established. The manuscript should better situate its work within the existing literature on LLM, explaining how it advances the state-of-the-art.

Minor Points of Review:

1. Clarity and Readability: the manuscript contains some typographical errors that need corrections for better readability. The structure of the manuscript could be improved by clearly delineating the sections on introductions, methods, and discussions.

Subheadings and bullet points can help in organizing the content more logically.

Conclusion: The study addresses an important issue for LLMs in supporting statistical consulting. Therefore, the manuscript requires substantial revisions. Furthermore, the novelty and contribution of the work are not established. Addressing the points raised above will be essential for improving the quality and impact of the manuscript.

Reviewer #2: The authors did an excellent job, and I have great appreciation for their dedication and hard work. Their commitment and effort are truly commendable

7. PLOS authors have the option to publish the peer review history of their article (what does this mean?). If published, this will include your full peer review and any attached files.

Reviewer #1: **Yes: **Wudneh Ketema Moges

Reviewer #2: **Yes: **Teklebirhan Abraha Gebrehiwot

---

## [Editor Report · Decision Letter 1]

23 Jul 2024

Exploring the Potential of Large language models for integration into an academic Statistical Consulting Service: The EXPOLS Study Protocol

PONE-D-24-24617R1

Dear Dr. Fichtner,

We’re pleased to inform you that your protocol has been judged scientifically suitable for publication and will be formally accepted for publication once it meets all outstanding technical requirements.

Within one week, you’ll receive an e-mail detailing the required amendments. When these have been addressed, you’ll receive a formal acceptance letter, and your protocol will be scheduled for publication.

If your institution or institutions have a press office, please notify them about your upcoming paper to help maximize its impact. If they’ll be preparing press materials, please inform our press team as soon as possible—no later than 48 hours after receiving the formal acceptance. Your protocol will remain under strict press embargo until 2 pm Eastern Time on the date of publication. For more information, please contact onepress@plos.org.

Kind regards,

Bekalu Tadesse Moges

Academic Editor

PLOS ONE
---

## [Editor Report · Acceptance letter]

21 Aug 2024

PONE-D-24-24617R1 

PLOS ONE

Dear Dr. Fichtner, 

I'm pleased to inform you that your manuscript has been deemed suitable for publication in PLOS ONE. Congratulations! Your manuscript is now being handed over to our production team.

Kind regards, 

on behalf of

Dr. Bekalu Tadesse Moges 

Academic Editor

PLOS ONE